# CMRSegTools: An open-source software enabling reproducible research in segmentation of acute myocardial infarct in CMR images

**William A. Romero R.**[1], **Magalie Viallon**[1]*, **Joël Spaltenstein**[2], **Lorena Petrusca**[1], **Olivier Bernard**[1], **Loïc Belle**[3], **Patrick Clarysse**[1], **Pierre Croisille**[1]

**1** Univ Lyon, INSA-Lyon, Université Claude Bernard Lyon 1, UJM-Saint Etienne, CNRS, Inserm, CREATIS UMR 5220, U1294, Saint Etienne, France, **2** Spaltenstein Natural Image, Geneva, Switzerland, **3** Department of Cardiology, Centre Hospitalier Annecy-Genevois, Annecy, France

* Magalie.Viallon@creatis.insa-lyon.fr

**Data Availability Statement:** In vivo images were obtained from the MIMI study (ClinicalTrials.gov Identifier: NCT01360242,217) and are available on

## Abstract

In the last decade, a large number of clinical trials have been deployed using Cardiac Magnetic Resonance (CMR) to evaluate cardioprotective strategies aiming at reducing the irreversible myocardial damage at the time of reperfusion. In these studies, segmentation and quantification of myocardial infarct lesion are often performed with a commercial software or an in-house closed-source code development thus creating a barrier for reproducible research. This paper introduces CMRSegTools: an open-source application software designed for the segmentation and quantification of myocardial infarct lesion enabling full access to state-of-the-art segmentation methods and parameters, easy integration of new algorithms and standardised results sharing. This post-processing tool has been implemented as a plug-in for the OsiriX/Horos DICOM viewer leveraging its database management functionalities and user interaction features to provide a bespoke tool for the analysis of cardiac MR images on large clinical cohorts. CMRSegTools includes, among others, user-assisted segmentation of the left-ventricle, semi- and automatic lesion segmentation methods, advanced statistical analysis and visualisation based on the American Heart Association 17-segment model. New segmentation methods can be integrated into the plug-in by developing components based on image processing and visualisation libraries such as ITK and VTK in C++ programming language. CMRSegTools allows the creation of training and testing data sets (labeled features such as lesion, microvascular obstruction and remote ROI) for supervised Machine Learning methods, and enables the comparative assessment of lesion segmentation methods via a single and integrated platform. The plug-in has been successfully used by several CMR imaging studies.

the Human Heart Project (http://humanheart-project.creatis.insa-lyon.fr) under research collaboration agreement. The source code is available at https://github.com/OpenCardiacMRISoftware/CMRSegTools.

**Funding:** This work has been performed within the framework of LABEX PRIMES 372 (ANR-11-LABX-0063) of Université de Lyon, within the program "Investissements d'Avenir" (ANR-11-IDEX-0007) and France Life Imaging (ANR-11-INBS-0006) operated by the French National Research Agency (ANR). The funders had no role in study design, data collection and analysis, decision to publish, or preparation of the manuscript.

**Competing interests:** The authors have declared that no competing interests exist.

# Introduction

Myocardial infarct size is a crucial parameter serving as a surrogate endpoint in clinical studies of new therapeutic interventions after Acute Myocardial Infarction (AMI). Therefore, considerable research engagement has been directed toward the automatic quantification of the infarct size as well as the size of the post-reperfusion no-reflow lesion based on CMR Late Gadolinium Enhanced (LGE) images [1–5]. These studies require reliable post-processing tools for the segmentation and quantification of the myocardial infarct lesion; in general, these tools are reported to be an in-house code development in a numeric computing environment (i.e. MATLAB) or a commercial application software. Not only do these tools restrict reproducibility of the studies but they can also prevent a full understanding of the use of computational methods. Because what is behind the mathematical model (constraints, parameters, etc.) is not clear, it is difficult to provide an objective assessment of lesion segmentation methods. For instance, a recent publication of Wu et al. [5] presents a general review of cardiac scar segmentation methods (including methods for non-LGE images). This review reports quantitative results across publications. The conclusion states the need of a public benchmarking of the methods as a fairer review of their performances. This is a challenge that requires tools for open collaboration in a growing culture focused on advancing methods by reproducible science [6, 7]. The neuroimaging research community is a clear example of this research and development methodology [8, 9].

In the field of Cardiac MRI, there is a gap between open-source software built within the scientific community and commercial applications. The issue is not the availability of computational methods for image processing in open-source packages *per se*, but the fact that these methods are not assembled in a suitable workflow for processing and carrying out statistical analysis on large clinical cohorts. A suitable workflow requires to select the structure of interest (myocardial segmentation), perform a computational method on the data from this structure, and analyse the output based on the American Heart Association (AHA) 17-segment model [10] (commonly named bullseye plot). Although it does not seem complex, it has been apparently embraced only by commercial applications which offer a custom-made workflow for myocardial segmentation and statistical analysis based on the AHA model. To provide an example, Table 1 presents an overview of well-known image processing tools in terms of two requirements for Cardiac MR image analysis: myocardial segmentation and statistical analysis following the standardised AHA 17-segment model. In general, a commercial software is a product designed to be marketed under a utilisation licence. The user must pay for a binary

**Table 1. Overview of available post-processing tools. Requirement 1 (R1): myocardial segmentation, requirement 2 (R2): statistical analysis and visualisation following the AHA 17-segment model.**

| Application Software | Description | R1 | R2 | Target Platform | Licence |
|---|---|---|---|---|---|
| 3D Slicer [13] | Subject-specific biomedical image processing and visualisation | No | No | Linux, Windows, macOS | Open-Source, BSD-style |
| ImageJ [14] | Multi-purpose scientific image processing and visualisation | No | No | Linux, Windows, macOS | Public Domain, BSD-2 |
| MeVisLab [15] | Workflow-based fast-prototyping | No | No | Linux, Windows, macOS | Proprietary, freeware |
| Medis Suite MR [16] | Cardiac MR image analysis | Yes | Yes | Windows | Commercial |
| Circle CVI42 [17] | Cardiac MR image analysis | Yes | Yes | Windows, macOS | Commercial |
| Segment CMR [18] | Cardiac MR image analysis | Yes | Yes | MATLAB on Linux, Windows, macOS | Segment open licence agreement |
| Caas MR Solutions [19] | Cardiac MR image analysis | Yes | Yes | Windows | Commercial |
| CMRSegTools | Cardiac MR image analysis | Yes | Yes | macOS | CeCILL |

version which must be used as in accordance with the licence directives. There is no access to the source code of the commercial product (closed-source). Freeware software is free of charge but closed-source. In open-source software, the user has access to the source code and the different licences (i.e. BSD, GNU GPL, etc.) provide the rights to copy, modify, redistribute the code, as well as build and release binary versions for commercial purposes [11]. Open-source software promotes open collaboration and research communities have been attracted to this innovation methodology [12]. Overall, open-source image processing software is not closely linked to the CMR concepts and practice. Therefore, a customised computer-based tool such as the one presented in this paper, could provide on-the-job support to ease analysis task performance on large clinical cohorts.

Moreover, a new demand has arisen with the advent of computational methods based on Machine Learning (ML). Using ML, the scientific community is now developing new solutions to solve the long-standing problem of a reliable quantification of infarct lesions. These approaches require annotated data sets which are generated and assessed by standard methods (i.e. signal threshold versus reference mean, full-width at half-maximum, etc.) [4, 5, 20, 21]. Consequently, there is a clear need for a unified reference of computational methods for lesion segmentation. This could be met with a common assessment platform that could easily be used by clinicians, radiologists and MRI researchers.

This paper presents CMRSegTools, a novel application software that allows users to benchmark new lesion segmentation methods as compared to state-of the-art methods in the field. CMRSegTools, includes customised tools for user-assisted myocardial segmentation; manual, semi- and automatic lesion segmentation which can be used in the generation of training data sets for Machine Learning (ML) strategies. This computer-based tool can be easily deployed on clinical infrastructures, closing the gap between analysis methods and their direct utilisation in clinical trials, as demonstrated in the studies [22–25].

## CMRSegTools plug-in

CMRSegTools is an application software specifically designed to ease myocardial segmentation, quantification and tissue characterisation on CMR images. CMRSegTools has been implemented as a plug-in for the widely available commercial OsiriX and open-source Horos DICOM viewers [26, 27] in order to take advantage of usability features and advanced visualisation tools (ROI outline tools, colour look-up tables, pixel interpolation algorithms for display, etc.) as well as the effortless deployment on a clinical or research infrastructure. Fig 1 shows the main components of the plug-in and its integration within the OsiriX/Horos environment.

The user-friendly Graphical User Interface (GUI) empowers the user to: segment the Left-Ventricle (LV); calculate infarct size, Microvascular Obstruction (MVO) lesion size, Endocardial Surface Length (ESL), and Endocardial Surface Area (ESA) amongst others. The real-time feedback updates the GUI with statistical information (number of pixels, minimum and maximum pixel value, mean value, standard deviation, etc.) of the pixels selected by the segmentation along with an interactive histogram. The image of interest is also dynamically updated (Epicardium and Endocardium contours, LV/RV junction landmark, region segments statistics in the 17-segment model, AHA [10]) according to the user interaction (Fig 2). Furthermore, CMRSegTools saves the workspace state, which means that it restores the GUI and image viewer to the state they were in when the plug-in was last used. The source code of the plug-in is available at https://github.com/OpenCardiacMRISoftware/CMRSegTools.

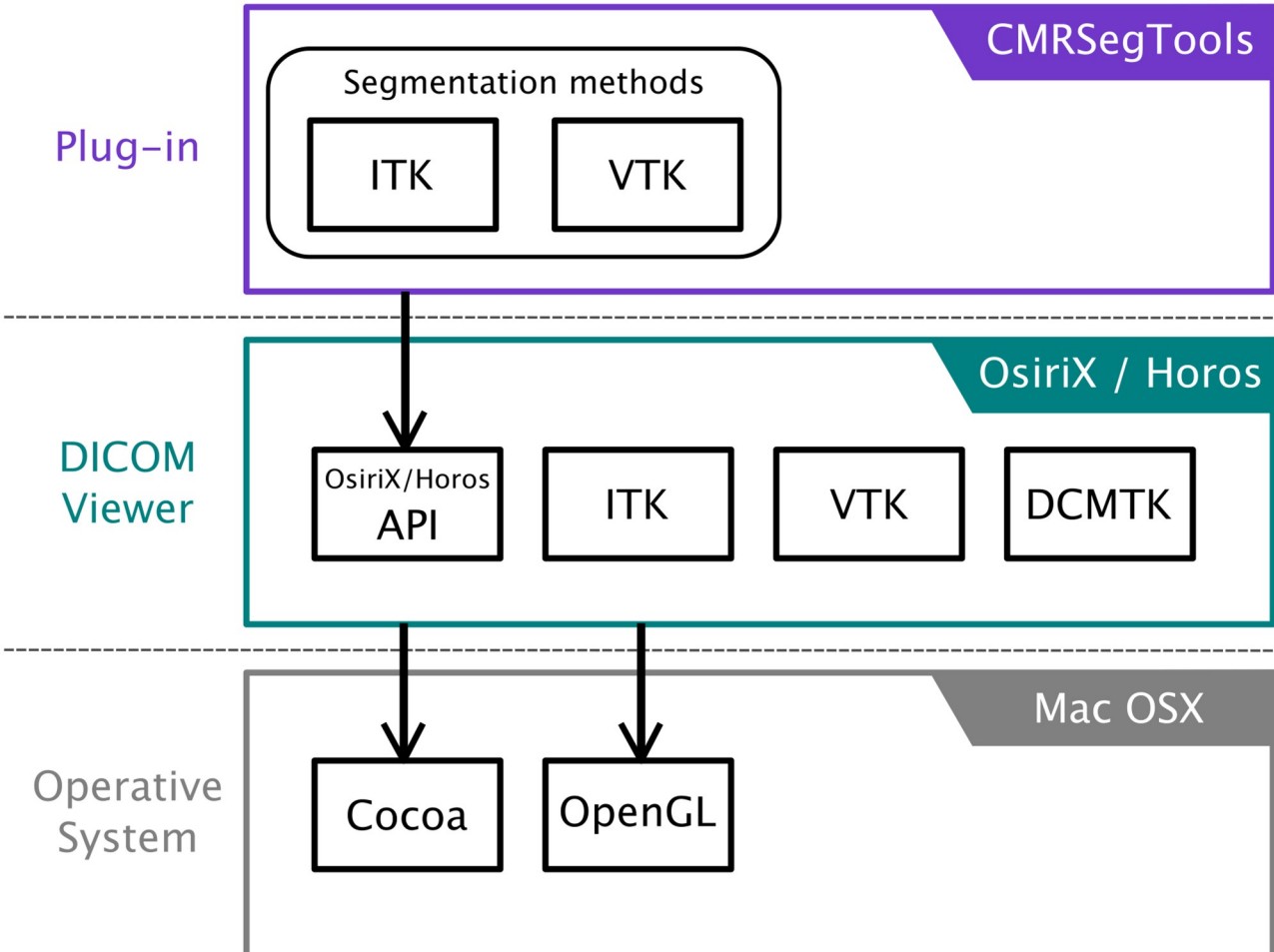

**Fig 1. CMRSegTools architecture diagram.** The plug-in has access to the DICOM viewer functionalities through the Application Program Interface (API) exposed by OsririX/Horos. The DICOM viewer runs on MacOS operating system. List of the frameworks and toolkits in each layer: ITK, Insight Toolkit; VTK, Visualisation Toolkit; DCMTK, DICOM Toolkit; Cocoa, Mac OSX Native AP; OpenGL, Computer Graphics API.

### Interoperability and extension

The CMRSegTools plug-in is independent of scanner vendor, sequence type and protocol. The OsiriX/Horos framework [26, 27] eliminates common requirements for data access (access to PACS), database management, transformation from the DICOM standard to final post-processing format, and data provenance; requirements which are critical in the development of clinical trials. In addition, the plug-in includes a functionality to import ROI from cvi42 (Circle Cardiovascular Imaging Inc., Calgary, Canada) file format.

This application software has been designed with the CMR lexicon in mind. All graphical elements generated by the plug-in (i.e. segmentation contours, set of pixels corresponding to the lesion, etc.) are tagged with keywords for identification within the OsiriX/Horos runtime environment. This enables the interaction with other plug-ins. For example, a specialised plug-in for myocardial segmentation (i.e. ML-based segmentation [21]) can export the segmentation output with the keywords *"CMRSegTools: Epicardium"* and *"CMRSegTools: Endocardium"* (ROI names can be set by the user in the OsiriX/Horos ROI manager). Therefore, this

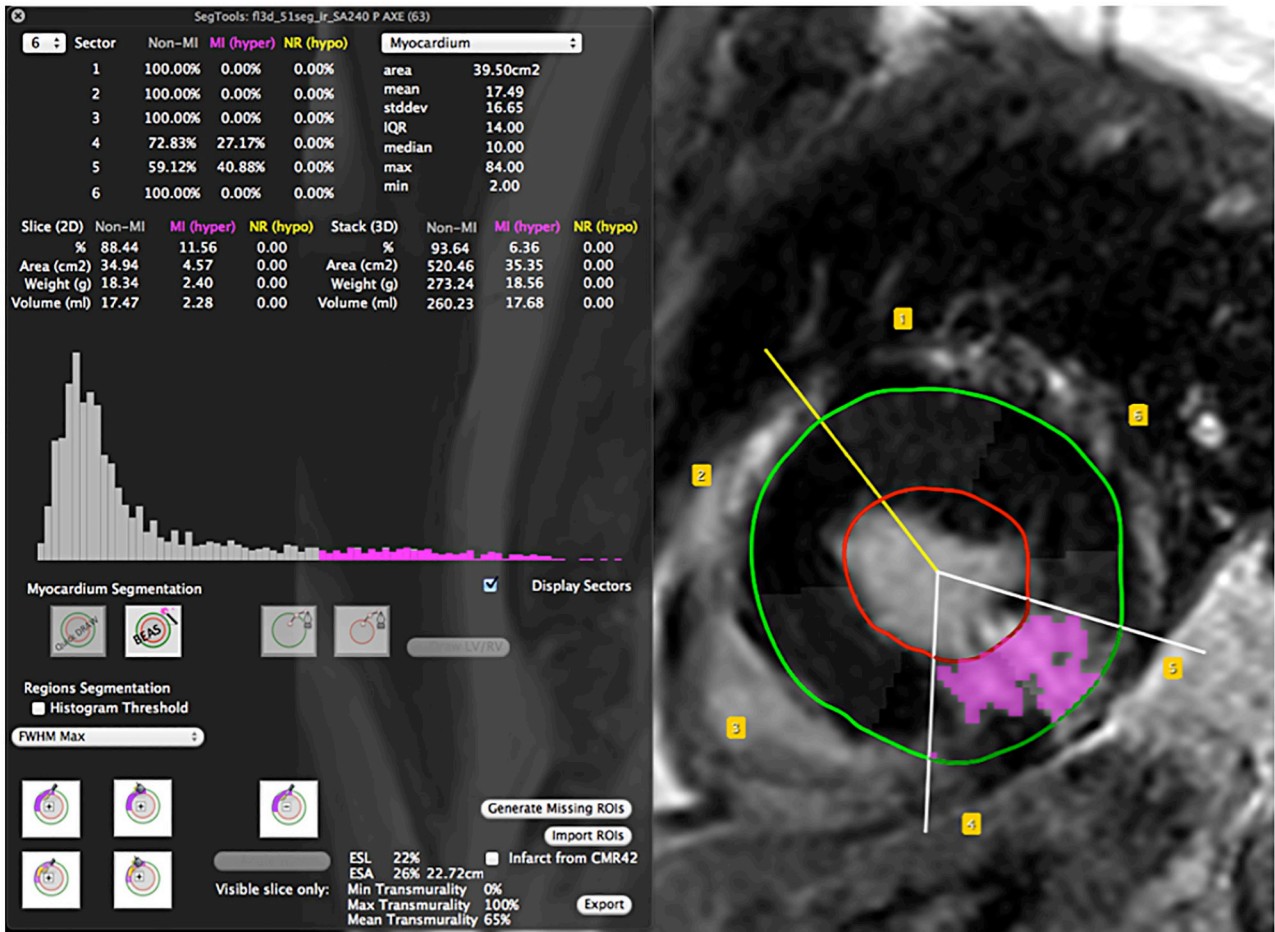

**Fig 2. CMRSegTools workspace.** The CMRSegTools GUI (left) shows the histogram highlighting the bins matching the lesion segmentation. The user can select the algorithms for the segmentation of the myocardium or some of its regions. The image viewer (right) displays the segmented area in magenta as well as the Epicardium (green) and Endocardium (red) contours, LV/RV junction landmark (yellow segment) and AHA region segments. The LGE image presented was obtained with a 3D IR GRE sequence.

segmentation can be identified by CMRSegTools. Inversely, another plug-in can start working with the CMRSegTools ROIs (*"CMRSegTools: Epicardium"*, *"CMRSegTools: Endocardium"*, *"CMRSegTools:Remote_stddev"*, etc.) by searching for these keywords in the OsiriX/Horos ROI manager.

Statistical results include the number of pixels, minimum and maximum pixel values, mean value, and standard deviation, among others of each ROI (lesion, MVO and remote) per segment, slice and for the entire myocardium. This information can be exported to an Excel file or as a delimited text file (comma-separated values) for a more comprehensive examination over different studies or cohorts of interest using application software for statistical analysis (e.g. pandas [28], R scripts [29] or any statistical package).

The main computational methods are written in C++ using the Insight Segmentation and Registration ToolKit (ITK) [30] and the Visualisation ToolKit (VTK) [31], which grant extensibility and interoperability in terms of new functionalities required, that is new image processing methods.

## Myocardial segmentation

The initialisation of the endocardial and epicardial contours is performed in 2 mouse clicks, and then automatically adjusted to the myocardium using the Boundary Enhanced Automated Surface (BEAS) algorithm [32, 33].

BEAS was originally developed for automatic segmentation of 3D cine-loop echocardiographic images to provide an efficient, fast and accurate solution for quantification of the main left ventricular volumetric indices used in clinical routine. In addition, the method was adapted for 3D+time CMR data sets (acquired by a cine steady state free precession sequence, SSFP) and benchmarked against the data sets available from the MICCAI 2009 Cardiac MR Left Ventricle Segmentation Challenge [34]. The benchmark demonstrated the technique to be robust, efficient and fast in terms of accuracy and computational load, which makes the BEAS algorithm suitable for a clinical practice [33].

The contours automatically determined by BEAS can be interactively corrected using OsiriX/Horos ROI edition functionalities such as the repulsor tool. Endocardial and epicardial contours can also be manually outlined. Moreover, the contours can be delineated on the basal, mid-cavity and apical slices and automatically propagated to the intermediate slices.

In order to remove pixels from the segmented myocardium close to the endocardial and epicardial contours, and avoid contamination from partial volume effect between myocardial tissue and blood (in the cavity) or air (at the myocardium to lung interface), an inner (epicardial) and outer (endocardial) offset in millimetres can be optionally set as a parameter of the segmentation method. These parameters can be configured by the user in the plug-in preferences.

## Lesion segmentation

Methods for measuring infarct size can be divided into visual assessment, manual planimetry, and voxel-based approaches. Visual assessment scores hyperenhancement on a 5-point grading scale on the AHA 17-segment model [10]. Manual planimetry involves a manual definition of the hyperenhanced regions of interest (ROI) across contiguous short-axis slices in order to calculate the lesion size [10, 35]. Both methods are inherently operator dependent, which makes them time consuming as they rely on a manual process. Consequently, they are unsuitable for clinical routine or the analysis of large cohorts. Voxel-based thresholding techniques start with a comparative measurement of hyperenhanced regions with a remote healthy region set as a reference (Signal Threshold versus Reference Mean, STRM) [36]. Threshold calculation can be based on a statistical measure such as the Full-Width at Half-Maximum (FWHM) [37, 38] or on histogram information as in the case of the Gaussian mixture model (GMM) approaches [39]. Depending on whether the methods require an initialisation parameter to calculate the threshold or not they are classified as semi-automatic or fully automatic. These methods are more time efficient and suitable for large scale processing.

The methods for the segmentation of the hyperenhanced region in CMRSegTools include:

- manual cut-off: the threshold is calculated as $T = \mu + c\sigma$ where $\mu$ is the mean of signal intensity in the remote and healthy myocardium and $\sigma$ its the standard deviation, $c$ is the parameter to set by the user (a positive integer usually between 2 and 10);

- manual histogram-based segmentation: the pixel intensity range is defined by two interactive cursors on the histogram;

- automatic cut-off: the threshold is calculated as $T = \frac{I_{max}}{2}$ where $I_{max}$ is the maximum intensity within the myocardium in the infarct zone (method known in the literature as Full-Width at Half-Maximum, FWHM);

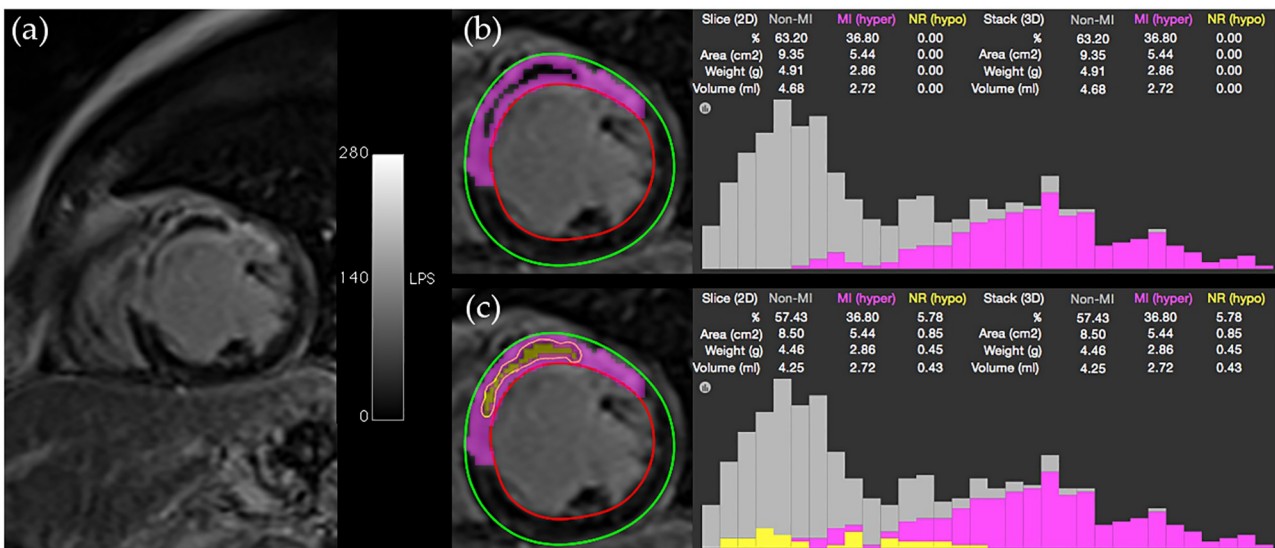

**Fig 3. CMRSegTools segmentation output.** (a) Image viewer before executing CMRSegTools; (b) Epicardium and endocardium contours from the myocardial segmentation, lesion segmentation (pixels in magenta colour), histogram and quantification statistics without MVO region; (c) Segmented regions, histogram and quantification statistics after defining the MVO region (contour and pixels in yellow colour).

- Gaussian mixture model: the threshold is defined as the intersection of a Rician-Gaussian mixture model fitted on the histogram [39];

- Hsu modified: this method is based on the feature analysis and combined thresholding (FACT) algorithm where an iterative refining process eliminates false-positives based on regional analysis [37];

- hidden Markov random field model with expectation-maximisation (HMRF-EM): this algorithm is founded on the method proposed in [40]. It labels pixels within the myocardium by calculating the set of parameters (mean and standard deviation) of the Probability Density Function (PDF) that defines each label (lesion and remote) [41].

These methods are described in detail in S1 Appendix.

## Manual definition of the MVO region

Only Hsu and HMRF-EM methods provide a segmentation of the MVO (or no-reflow) regions. The MVO region can be selected (or edited) manually after segmentation of the infarct by using a pixel selection tool (brush) or region growing functionality. If there is a MVO region, this new class of tissue is automatically added to the infarct size, with corresponding pixels being automatically displayed in yellow (default color on the GUI for highlighting the MVO region). Similarly, if MVO pixels are categorised as false positive and highlighted in yellow by Hsu or HMRF-EM methods, they can be manually removed. MVO size and statistics (number of pixels, minimum and maximum pixel value, mean value, standard deviation, etc.) are calculated (Fig 3).

## Transmural extent, endocardial surface length and endocardial surface area

An interactive 2D/3D wiper tool (white segments in Fig 2) allows the user to manually identify the lateral edges of myocardial infarct in order to determine the Endocardial Surface Length

(ESL) of the infarct (in percent of the endocardial contour perimeter) [42]. When multiple slices are evaluated (wipers location is propagated across slices), the Endocardial Surface Area (ESA) is automatically calculated. This functionality also intrinsically allows the user to clean up the segmentation results a posteriori and provides a transmural extent estimation.

## Testing data

A series of synthetic images emulating myocardial infarct lesions were generated with arbitrary size, transmural extent, percent Signal Enhancement (%SE) and CNR. In order to test the influence of the noise level on the results, several synthetic levels of noise were computed to reproduce what was previously measured in vivo using various CMR sequences in patients [43]. To achieve a realistic model of the cavity, myocardium, infarct size and shape, and signal intensity in each compartment, the numeric images were built using multi-slice 3D data sets acquired on patients. Synthetic images were generated with a resolution similar to the one used in clinical acquisitions [41].

In vivo images from the MIMI study (ClinicalTrials.gov Identifier: NCT01360242, [44]) were used in this study to validate the plug-in functionalities on real clinical data. The MIMI (Minimalist Immediate Mechanical Intervention) study was a multicentric randomised trial aiming at comparing immediate stenting and 24–48h delayed stenting in patients treated with primary percutaneous coronary intervention (PCI); the MR protocol included T1, T2, T2*, Cine and 3D/2D Late Gadolinium Enhancement (LGE) scans. The data set is available through the Human Heart Project [45] under research collaboration agreement.

## Results

The validation of the implemented methods within CMRSegTools includes a functional validation of the myocardial segmentation method; verification of the lesion segmentation output on synthetic data, and an assessment of the automated segmentation functionalities based on a reference made by a radiology expert.

### Myocardial contour segmentation with BEAS

An example of the CMRSegTools LV segmentation output is presented in Fig 4. These results show the variation of the segmented contours across different image types (LGE, EGE and cine images) as a consequence of the image contrast difference and myocardial mass.

Although the BEAS algorithm was originally developed for segmentation of echocardiographic images, the method has shown very good performance on images with homogeneous myocardium wall (well-defined ring structure) such as cardiac cine MRI based on balanced SSFP contrast. As this method is based on active contours principle [46, 47], the segmentation is affected by the heterogeneity of the myocardium wall with multiple tissue classes (lesion, MVO and remote healthy pixels). These tissue classes have an impact on the energy function that penalises the deviation to the initial model (LV template presented in Fig 4a and 4d), which means that the user may have to manually correct the LV segmentation contours.

### Validation on synthetic data

The numerical phantom generation from real clinical data (2D and 3D) is illustrated in Fig 5 and a performance benchmark across segmentation methods is presented in Fig 6. Fig 6 shows the relative and absolute error in the calculation of the lesion size as a function of the CNR. Absolute relative error is plotted on logarithmic scale to highlight the results of 2-SD, 3-SD and 5-SD for $CNR > 5$. Overall, the accuracy improves as the CNR increases. The

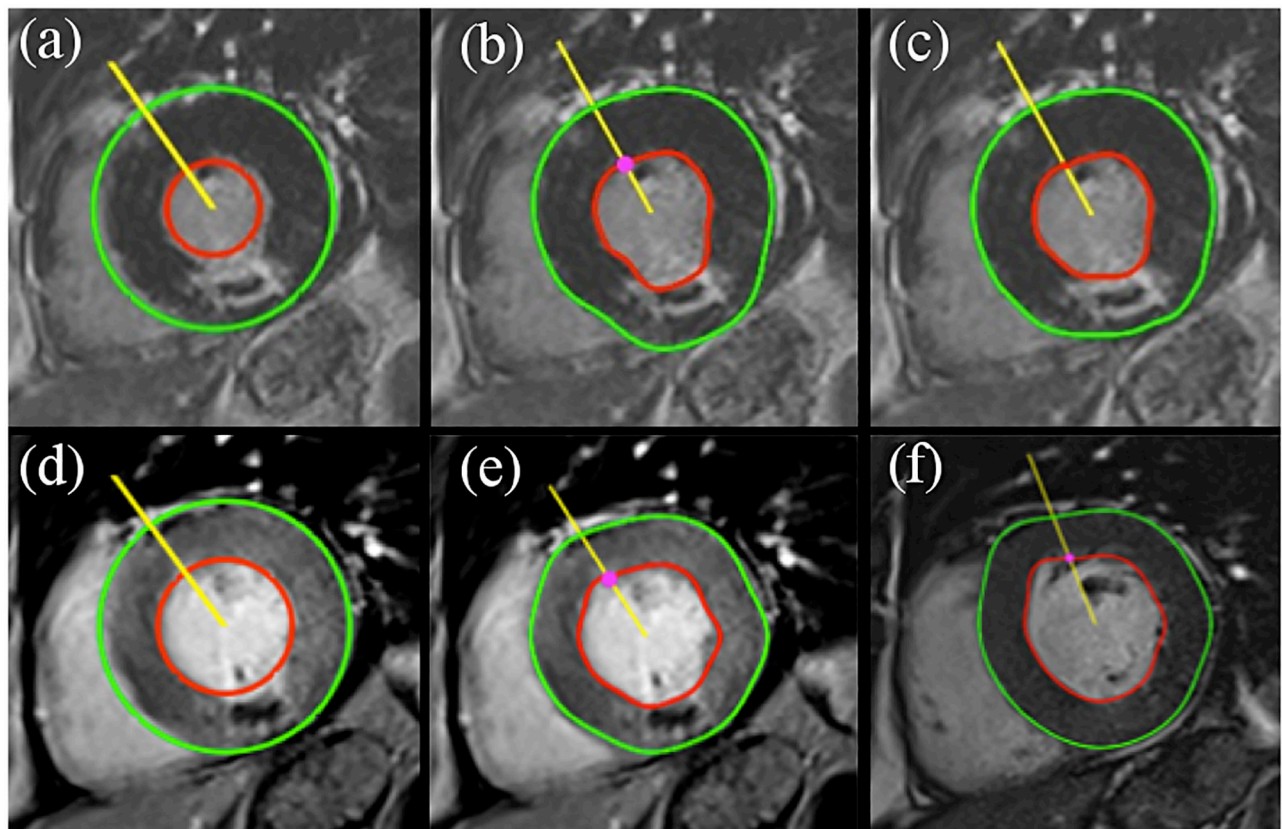

**Fig 4. Performance of BEAS segmentation on images with different contrast type.** (a) User initialisation on the LGE image; (b) first and (c) second iteration of the automatic segmentation based on (a). (d) User initialisation on the EGE image and (e) segmentation output. (f) Segmentation output on cine bSSFP. All images were taken at the same slice location and from the same patient at different breath-hold scans.

segmentation by STRM (n-SD) methods is strongly determined by the remote region and CNR. Lower CNR ($CNR < 6$) leads to an under-estimation (less pixels categorised as MI) by 2-SD, 3-SD,5-SD and FWHM Region methods, and to an over-estimation (more pixels categorised as MI) by FWHM Max, GMM, Hsu and HMRF-EM methods. This is due to the incorrect categorisation of the infarct edges or some noisy pixels. On a real image coil sensitivity variation, cardiac motion and partial-volume effects may generate this kind of intermediate SI pixels that may be included within the n-SD constraint or statistical model. However, at lower CNR scenarios, a lower cut-off provides a more accurate infarct segmentation with an error less than 3.8%. Lower CNR impacts the performance of the GMM method reaching about 60% of mean relative error. The best performance corresponds to STRM with 5-SD, FWHM, Hsu and HMRF-EM providing a very low margin of error starting at $CNR > 3$.

## Radiology expert segmentation vs automated segmentation methods

An example of the manual segmentation made by an expert radiologist (P.C. with 20 years of experience in cardiovascular imaging) against each one of the methods within CMRSegTools on a patient with MVO is presented in Figs 7 and 8. The relative error between methods output and expert segmentation of a ROI was calculated as $Error_{ROI} = \frac{Area_{ROI}(method) - Area_{ROI}(expert)}{Area_{ROI}(expert)}$

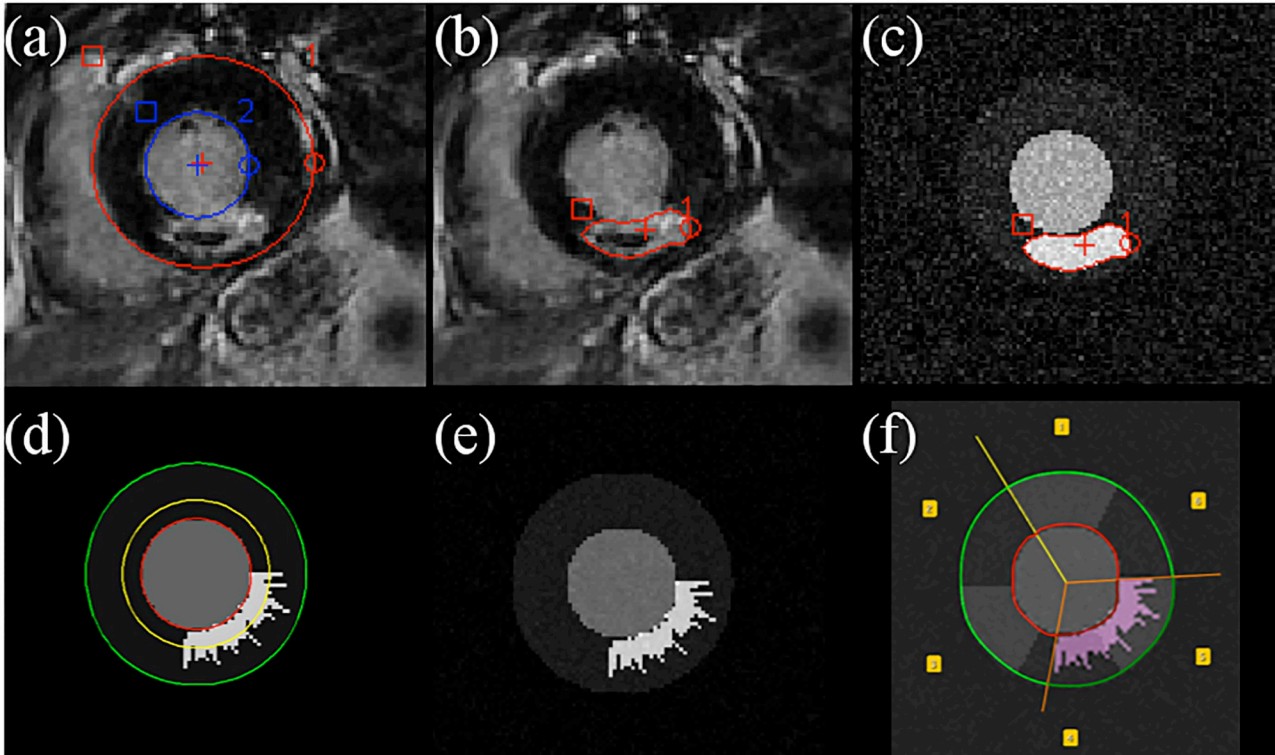

**Fig 5.** Example of numerical phantom generation (c), (d), (e) and (f) from a real clinical data set (a) and (b). Images were acquired with a 3D-IR-GRE sequence on a patient with a myocardial infarct. (a) Traced endocardial and epicardial contours on one slice (2D), (b) manual lesion segmentation, (c) output synthetic image (2D). The figures (d), (e) and (f) illustrate transmural extent and ESL calculation on a numerically generated infarct with known size and variable transmural extent around a user-defined mean value. (d) Red, green and yellow contours are respectively the endocardial and epicardial borders, and the mean simulated transmural extent; (e) final synthetic image. (f) CMRSegTools segmentation output (HMRF-EM); magenta pixels on the synthetic image correspond to those classified as myocardial infarct on the numerical phantom. The orange wipers delineate the ESL.

## Discussion

Although a considerable research has been conducted to develop and assess computational methods for the quantification of lesion size in CMR, to date and to the best of our knowledge, there has been no initiative to provide a unified reference for lesion segmentation methods within a common assessment platform. The baseline of existing publications reporting infarct lesion sizing is usually an offline analysis based on in-house (closed-source) implementations of conventional or, less often, innovative segmentation algorithms. Following this approach it is difficult to make an objective evaluation of existing methods. The assessment would only be performed when a medical device vendor releases the methods in a commercial software product. In this scenario, the benchmarking of computational methods implemented by research groups and commercial solutions is biased by the vendor constraints for the algorithms. Since manufacturers are under regulatory restrictions, the integration of the latest hardware and software on their platforms typically requires long production time, often resulting in regrettable delays in early testing of new, advanced and potentially valuable algorithms. CMRSegTools is therefore a contribution to enable reproducible research by assembling the most used methods for infarct segmentation in a suitable workflow executed in a single and widely available environment: Osirix/Horos DICOM viewers. This platform was chosen because it provides an Application Programming Interface (API) that grants access to the data management, user interaction and visualisation functionalities. It was also chosen because it has an open-source

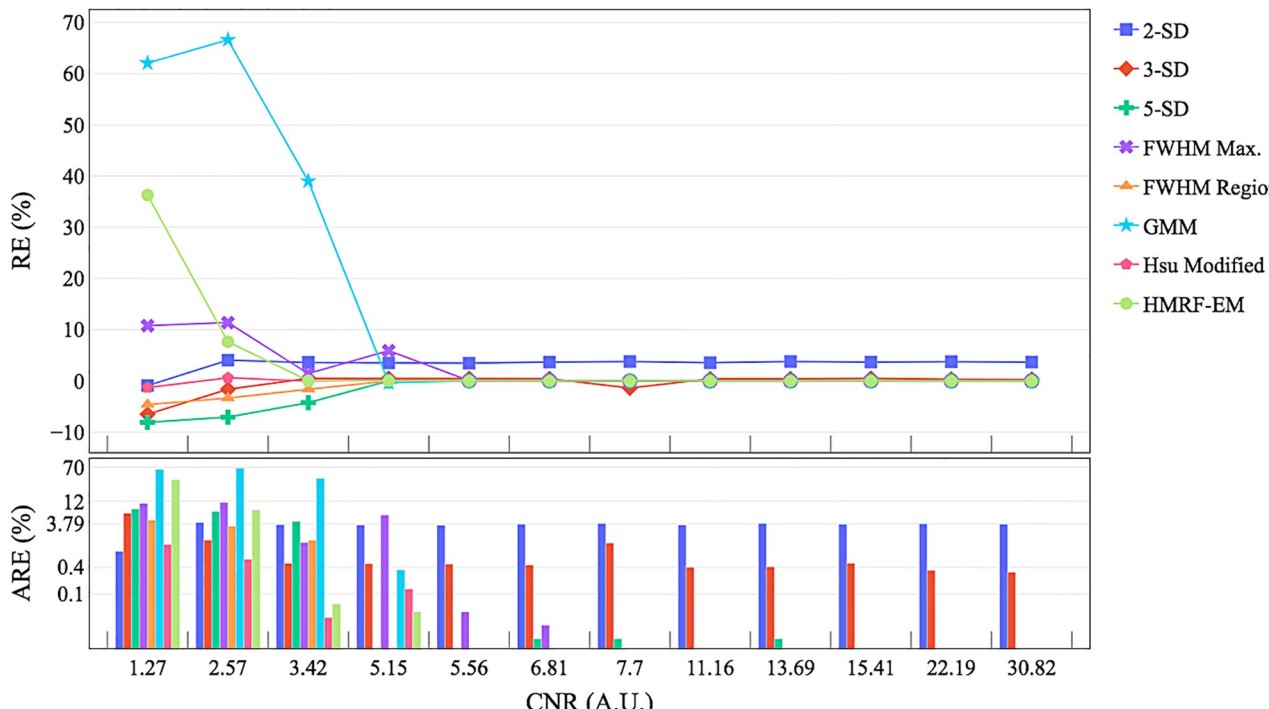

**Fig 6. Performance of the lesion segmentation methods across different Contrast-to-Noise Ratio (CNR) scenarios.** The curves show the relative error (RE) and bars the absolute relative error (ARE, logarithmic scale) for each segmentation method. The CNR was calculated as the ratio between the SI difference of the infarct and healthy myocardium over the standard deviation of the SI in the myocardium.

licence. However, OsiriX/Horos is a macOS-only software, which limits its deployment on platforms based on a different operating system.

One of the contributions to the infarct segmentation workflow is the implementation of a semi-automatic method for the LV segmentation based on active contours [32, 33]. The BEAS algorithm utilises a LV template provided by the user in order to automatically fit the template to the epicardium and endocardium boundaries. Although the outcome of the BEAS segmentation may be affected by the different tissue classes in the myocardial wall, this functionality enhances automation capabilities for clinical routine or the analysis of large cohorts as demonstrated in [33]. A reliable segmentation of the myocardium is important as the edge of the endocardium may constitute up to 50% of the infarct perimeter; this means that a wrong determination of the endocardial infarct border is a large source of variability in final infarct size measurements. At present, there are no automated algorithms that can reliably distinguish the bright LV cavity from the bright endocardial border of the infarct. This challenge is being addressed by new segmentation methods (including non-LGE images) based on ML [5, 21, 48, 49].

To validate the correspondence between the mathematical model (S1 Appendix) and the code implementation, the correctness of the computational methods has been verified with synthetic data. This also allowed the benchmarking of infarct segmentation methods (Fig 6) on synthetic data. The main lesson emerging from the analysis is that STRM (n-SD) methods with $n < 5$ and GMM are easily influenced by CNR. At lower CNR conditions the outcomes from these methods are unreliable, making them unsuitable for real clinical conditions. When $CNR > 5$, 2-SD and 3-SD are insensible to CNR. The HMRF-EM has shown a good

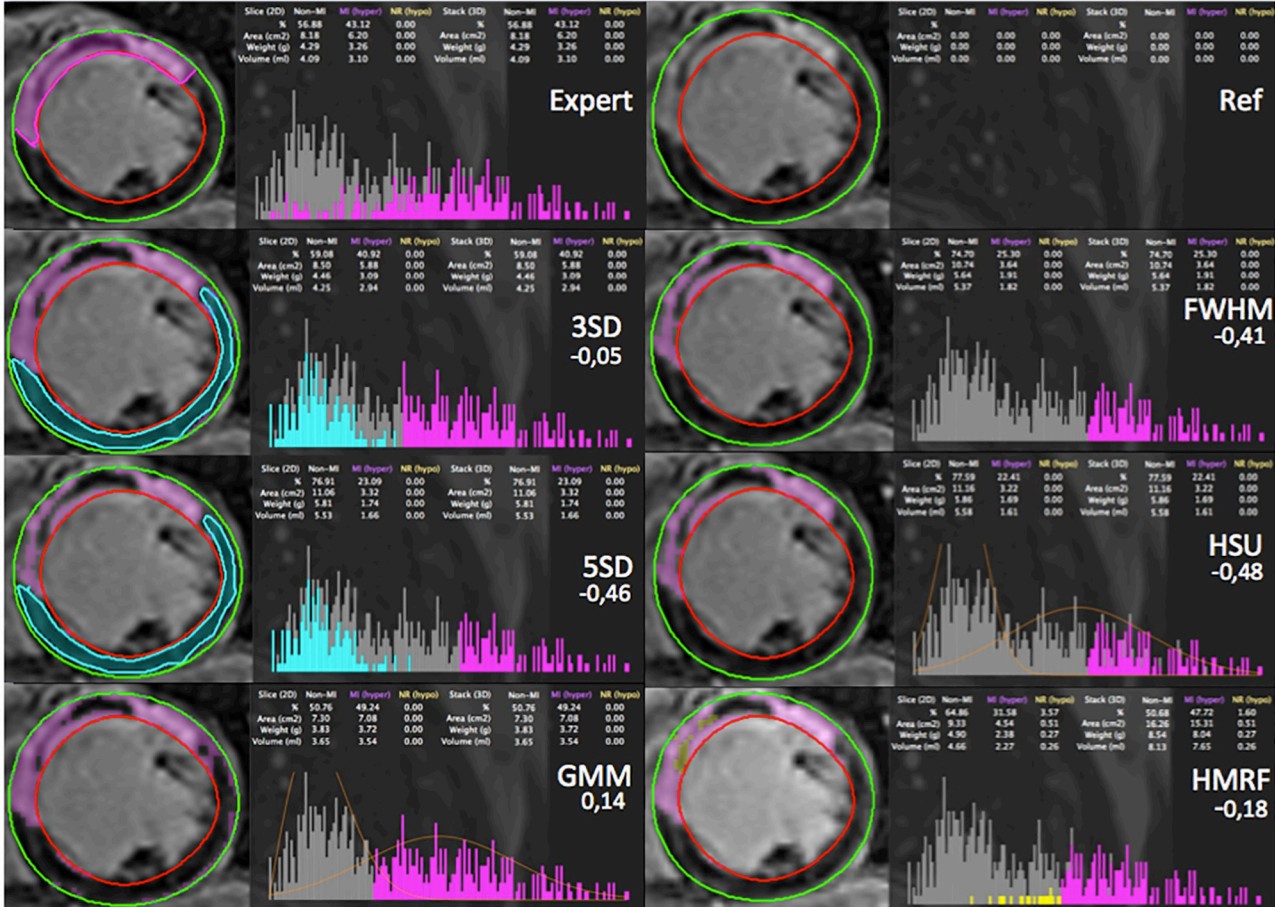

**Fig 7. Comparison between the manual segmentation made by an expert (without the definition of the MVO region), and the segmentation made by each of the methods within CMRSegTools.** For each segmentation, the figure shows: epicardium (green) and endocardium (red) contours from the myocardial segmentation, lesion segmentation (magenta), remote healthy myocardium (turquoise), no-reflow region by HMRF (yellow), histogram, quantification statistics and relative error.

performance in most of the different scenarios ($CNR > 3$). A comparison between expert and automated segmentation methods is presented in Figs 7 and 8 as an example of the output of the current version. CMRSegTools has been tested and used successfully in several studies [22–25].

An extended assessment or benchmarking of infarct segmentation methods is beyond the scope of this paper. The goal is to provide a computer-based tool which includes well-known lesion segmentation methods (S1 Appendix) in order to empower research scientists to reproduce results and assess comparatively new methods. In this direction, CMRSegTools source code has been released under the CeCILL licence (a French free software license compatible with GPL) that grants users the right to copy, modify, and distribute new versions of the plug-in.

Machine learning approaches for SI classification would require reference data that can be applied to a learning process [50]. With CMRSegTools, it is possible to generate labeled features (infarct, MVO and remote ROIs) extracted from the data and the expected quantification based on reference segmentation methods. These specific features are the main input of supervised learning approaches. As shown in Figs 7 and 8, outcomes from ML methods can be easily

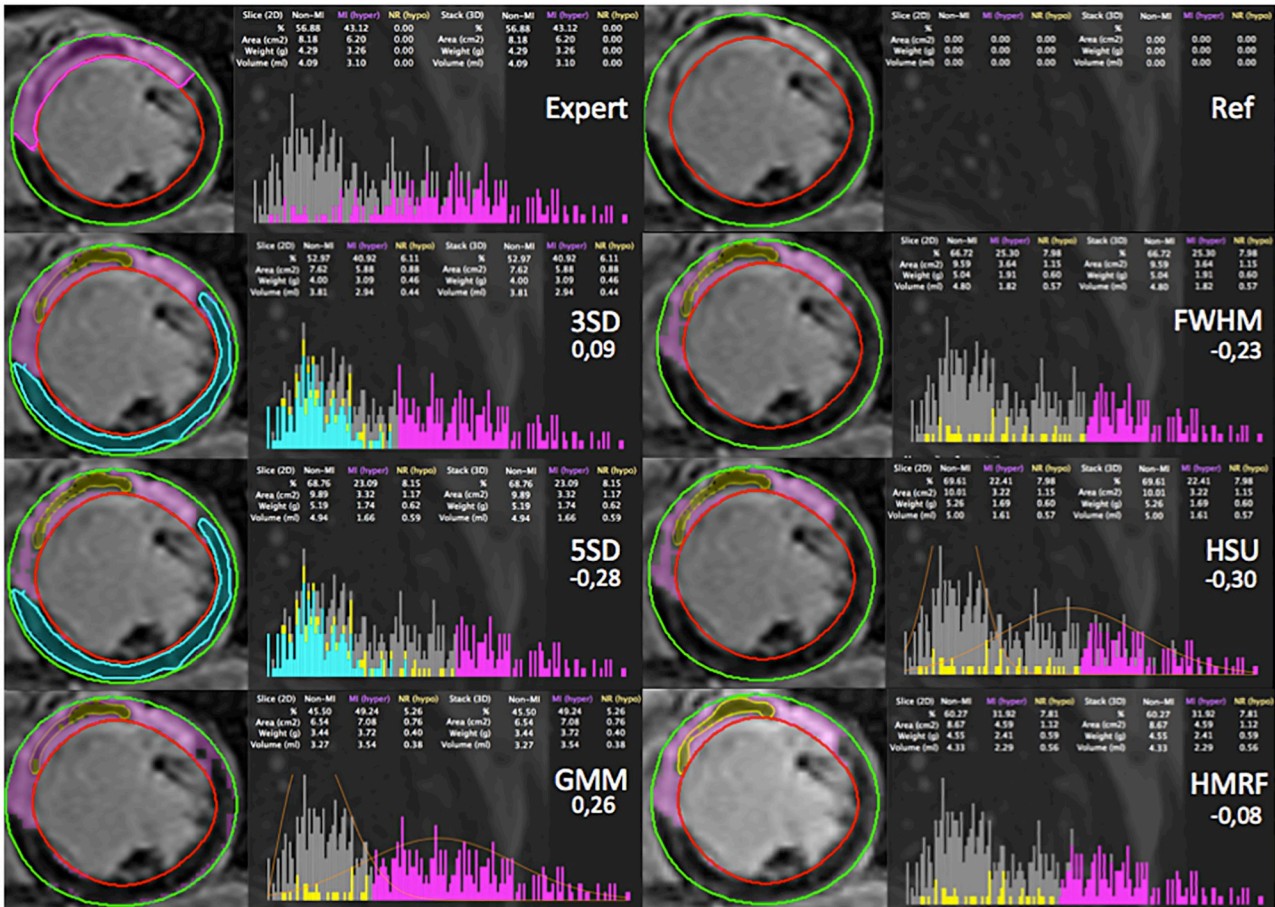

**Fig 8. Comparison between the manual segmentation made by an expert (including a manual definition of the MVO region), and the segmentation made by each of the methods within CMRSegTools.** For each segmentation, the figure shows: epicardium (green) and endocardium (red) contours from the myocardial segmentation, lesion segmentation (magenta), remote healthy myocardium (turquoise), no-reflow region (yellow), histogram, quantification statistics and relative error.

compared to manual, semi- or full-automated state-of-the-art methods. Additionally, errors performed by ML methods may be labeled as well in order to improve the accuracy, specially in strategies involving active learning.

The analysis scenario presented in this paper (methods benchmarking and expert vs computational methods assessment) can be replicated by research scientists working in CMR on their own data or by downloading the testing data used for this work from the Human Heart Project [45]. This is particularly interesting as it allows reproducible research in this field.

## Conclusions

CMRSegTools is an application software for the comparative assessment of infarct segmentation methods applied to CMR images via a single and integrated platform (widely available Osirix and Horos DICOM viewers). This helps to improve the reproducibility of post-processing methods in clinical studies. This application software works on native DICOM images; the OsiriX/Horos functionalities enable direct connection to PACS making the plug-in independent of scanner vendor, imaging sequence and protocol. Interoperability functionalities allow

importing data from external software such as ROI created in other application software (e.g. cvi42). New segmentation methods can be easily integrated into the plug-in by developing components based on well-known image processing and visualisation libraries such as ITK and VTK in C++ programming language. By using CMRSegTools, it is possible to create training and testing data sets that can be used in Machine Learning approaches. In addition, statistical measurements can be exported for further examination in specialised data analysis tools.

## Supporting information

**S1 Appendix. Lesion segmentation methods in CMRSegTools.**
(PDF)

## Acknowledgments

The authors would like to thank Alessandro Volz, M.Sc. (Medical imaging software developer. Bellinzona, Switzerland.) for his valuable contributions to this project.

## Author Contributions

**Conceptualization:** Magalie Viallon, Pierre Croisille.

**Data curation:** Magalie Viallon, Loïc Belle, Pierre Croisille.

**Formal analysis:** Magalie Viallon.

**Funding acquisition:** Loïc Belle, Pierre Croisille.

**Methodology:** Magalie Viallon, Patrick Clarysse, Pierre Croisille.

**Software:** William A. Romero R., Joël Spaltenstein, Olivier Bernard.

**Validation:** William A. Romero R., Magalie Viallon, Joël Spaltenstein, Lorena Petrusca, Olivier Bernard, Loïc Belle, Patrick Clarysse, Pierre Croisille.

**Writing – original draft:** William A. Romero R., Magalie Viallon.

**Writing – review & editing:** William A. Romero R., Magalie Viallon, Joël Spaltenstein, Lorena Petrusca, Olivier Bernard, Loïc Belle, Patrick Clarysse, Pierre Croisille.

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
