## [Decision Letter · Decision Letter 0]

21 Apr 2022

PONE-D-22-07768CMRSegTools: an open-source software enabling reproducible research in segmentation of acute myocardial infarct in CMR imagesPLOS ONE

Dear Dr. Romero R.,

Thank you for submitting your manuscript to PLOS ONE. After careful consideration, we feel that it has merit but does not fully meet PLOS ONE’s publication criteria as it currently stands. Therefore, we invite you to submit a revised version of the manuscript that addresses the points raised during the review process.

We look forward to receiving your revised manuscript.

Kind regards,

Kumaradevan Punithakumar

Academic Editor

PLOS ONE

Journal Requirements:

3. We note that you have stated that you will provide repository information for your data at acceptance. Should your manuscript be accepted for publication, we will hold it until you provide the relevant accession numbers or DOIs necessary to access your data. If you wish to make changes to your Data Availability statement, please describe these changes in your cover letter and we will update your Data Availability statement to reflect the information you 

Reviewers' comments:

Reviewer's Responses to Questions

**Comments to the Author**

1. Is the manuscript technically sound, and do the data support the conclusions?

Reviewer #1: Partly

Reviewer #2: Yes

2. Has the statistical analysis been performed appropriately and rigorously? 

Reviewer #1: N/A

Reviewer #2: Yes

3. Have the authors made all data underlying the findings in their manuscript fully available?

Reviewer #1: Yes

Reviewer #2: Yes

4. Is the manuscript presented in an intelligible fashion and written in standard English?

Reviewer #1: Yes

Reviewer #2: Yes

5. Review Comments to the Author

Reviewer #1: The paper presents an open-source software for segmentation of CMR-LGE images.

Quantitative LGE analysis is not strictly required by current CMR guidelines in the clinical setting (J Schulz-Menger JCMR 2020 10.1186/s12968-020-00610-6), but it is needed in clinical and translational research. A large number of LGE segmentation algorithms was proposed (Wu Front. Physiol 2021 | 10.3389/fphys.2021.709230), few of them established in the clinical practice.

About available software, several certified, commercial tools for CMR-LGE analysis are available at high cost from several companies. Some groups developed software for internal use but in my knowledge the only available free tools is the Segment software from Medvisio (https://medviso.com/cmr/) running by a free available Matlab VM on about all SO platforms. Segment is free usable if proper referencing is assured (https://medviso.com/how-to-refer/) but source code is not available, so the software cannot be adapted to user needs.

Hence, the introduction of on open-source software is important and may represent an important improvement in the field. I tried the demo version in Horos and it seems to work well. Basic video tutorials are provided, maybe a manual would be also useful.

Collaboration terms required for the use in scientific publication of the software (likely the only possible use as the software is of course not certified) are proper referencing as usually requested in the field and mandatory co-authorship of developers. The last request is not usual in open-source software sharing as co-authorship is usually applied id a specific developing work is done for a particular project. I guess that the GNU/CeCILL license doesn’t allow the developer to add further limitations with respect to the license itself. Is also not clear how this “co-authorship license” should apply if thanks to the open-source approach the software will be improved or modified by other developers external to the starting team. The “co-authorship license” constraint, if maintained, should be declared in the paper as well as it represents a significant limitation.

Validation of the software is mostly anecdotic as single cases as presented without quantitative assessment. Is strongly advisable to perform a comparison with a validated software or manual analysis by an expert user on a reference data set including a reasonable number of cases, as MICCAI challenges or similar.

Table 1 compares general-use software (that of course does not match R1/R2 requirements) with CMR specific ones that of course match both as required by SCMR guidelines. Research/Commercial Segment software should be inserted in the table together with other commercial software as Pie CAAS (https://www.piemedicalimaging.com/product/mr-solutions/tissue-characterisation). The compatibility with different OS should be indicated, as one limitation of the proposed software is the requirement of MacOS system.

Reviewer #2: This study presents a open-source software for myocardial infarction segmentation in CMR images. There are some concerns:

- It is better to brief introduce the licenses shown in Table 1 for clarifying the difference in different software.

- Is there any comparison of BEAS with other myocardial contour segmentation ? If not, please explain the reason.

- It is better to cite more studies in cardiac and myocardial infarction segmentation. 1) Direct delineation of myocardial infarction without contrast agents using a joint motion feature learning architecture, Medical Image Analysis; 2) Multi-level semantic adaptation for few-shot segmentation on cardiac image sequences, Medical Image Analysis.

- Please briefly discuss the limitation of this software

- A lot of grammatical errors.

- Please present a figure to show the modules and their connections in this software.

6. PLOS authors have the option to publish the peer review history of their article (what does this mean?). If published, this will include your full peer review and any attached files.

Reviewer #1: No

Reviewer #2: No

---

## [Author Response · Author response to Decision Letter 0]

28 Jun 2022

Reviewer #1

1. The paper presents an open-source software for segmentation of

CMR-LGE images. Quantitative LGE analysis is not strictly required by

current CMR guidelines in the clinical setting (J Schulz-Menger JCMR

2020 10.1186/s12968-020-00610-6), but it is needed in clinical and

translational research. A large number of LGE segmentation algorithms

was proposed (Wu Front. Physiol 2021 | 10.3389/fphys.2021.709230), few

of them established in the clinical practice.

 We thank the reviewer for the observation and references. We have

 included the latter reference in our bibliographical revision and

 discussion.

 The following paragraph has been added to the Introduction:

 For instance, a recent publication of Wu et al. [5] presents a general

 review of cardiac scar segmentation methods (including methods for

 non-LGE images). This review reports quantitative results across

 publications. The conclusion states the need of a public benchmarking

 of the methods as a fairer review of their performances. This is a

 challenge that requires tools for open collaboration in a growing

 culture focused on advancing methods by reproducible science [6,

 7]. The neuroimaging research community is a clear example of this

 research and development methodology [8, 9].

 The following sentence has been also added to the Introduction:

 These approaches require annotated data sets which are generated and

 assessed by standard methods (i.e. signal threshold versus reference

 mean, full-width at half-maximum, etc.) [4, 5, 19, 20]

 The following reference has been added:

 5. Wu, Yinzhe, Zeyu Tang, Binghuan Li, David Firmin, and Guang

 Yang. “Recent Advances in Fibrosis and Scar Segmentation From Cardiac

 MRI: A State-of-the-Art Review and Future Perspectives.” Frontiers in

 Physiology 12 (2021).

2. About available software, several certified, commercial tools for

CMR-LGE analysis are available at high cost from several

companies. Some groups developed software for internal use but in my

knowledge the only available free tools is the Segment software from

Medvisio(https://medviso.com/cmr/) running by a free available Matlab

VM on about all SO platforms. Segment is free usable if proper

referencing is assured (https://medviso.com/how-to-refer/) but source

code is not available, so the software cannot be adapted to user

needs.

Hence, the introduction of on open-source software is important and

may represent an important improvement in the field. I tried the demo

version in Horos and it seems to work well. Basic video tutorials are

provided, maybe a manual would be also useful.

Collaboration terms required for the use in scientific publication of

the software (likely the only possible use as the software is of

course not certified) are proper referencing as usually requested in

the field and mandatory co-authorship of developers. The last request

is not usual in open-source software sharing as co-authorship is

usually applied id a specific developing work is done for a particular

project. I guess that the GNU/CeCILL license doesn’t allow the

developer to add further limitations with respect to the license

itself. Is also not clear how this “co-authorship license” should

apply if thanks to the open-source approach the software will be

improved or modified by other developers external to the starting

team. The “co-authorship license” constraint, if maintained, should be

declared in the paper as well as it represents a significant

limitation.

 We thank the reviewer for testing our application software and

 pointing out the potential limitations of the use of CMRSegTools under

 the current “co-authorship licence”.

 Before releasing CMRSegTools as an open-source software, we provided a

 binary version of the plug-in as well as scientific expertise in

 applied computing in Cardiac MRI and software customisation to the

 particular needs of the users. This makes us active collaborators in

 the research projects using the application software. In this context,

 the co-authorship agreement has been working among research teams

 using CMRSegTools.

 Now our interest is to join efforts through an open

 collaboration. Releasing the source code of CMRSegTools together with

 this publication is our first step. The GNU/CeCILL is the French free

 software license, compatible with the GNU GPL, the license grants

 access to the source code and the rights to copy, modify and

 redistribute the code.

 We have also included Segment CMR (https://medviso.com/cmr) and Cass

 MR Solutions (https://www.piemedicalimaging.com/product/mr-solutions)

 in our software comparison.

 The sentence below has been added to the Discussion section:

 In this direction, CMRSegTools source code has been released under

 CeCILL licence (French free software license compatible with GPL) that

 grants users the right to copy, modify, and distribute new versions of

 the plug-in.

3. Validation of the software is mostly anecdotic as single cases as

presented without quantitative assessment. Is strongly advisable to

perform a comparison with a validated software or manual analysis by

an expert user on a reference data set including a reasonable number

of cases, as MICCAI challenges or similar.

 We thank the reviewer for this observation. The software has already

 been tested before this open-source release. All implemented methods

 have been evaluated in the publications cited in appendix S1.

 The following paragraph has been added to the CMRSegTools plug-in

 section:

 BEAS was originally developed for automatic segmentation of 3D

 cine-loop echocardiographic images to provide an efficient, fast and

 accurate solution for quantification of the main left ventricular

 volumetric indices used in clinical routine. In addition, the method

 was adapted for 3D+time CMR data sets (acquired by a cine steady state

 free precession sequence, SSFP) and benchmarked against the data sets

 available from the MICCAI 2009 Cardiac MR Left Ventricle Segmentation

 Challenge [33]. The benchmark demonstrated the technique to be robust,

 efficient and fast in terms of accuracy and computational load, which

 makes the BEAS algorithm suitable for a clinical practice [32].

 And in the Discussion section:

 To validate the correspondence between the mathematical model (S1

 appendix) and the code implementation, the correctness of the

 computational methods has been verified with synthetic data.

 An extended assessment or benchmarking of infarct segmentation methods

 is beyond the scope of this paper. The goal is to provide a

 computer-based tool which includes well-known lesion segmentation

 methods (S1 appendix) in order to empower research scientists to

 reproduce results and assess comparatively new methods. In this

 direction, CMRSegTools source code has been released under CeCILL

 licence (French free software license compatible with GPL) that grants

 users the right to copy, modify, and distribute new versions of the

 plug-in.

4. Table 1 compares general-use software (that of course does not

match R1/R2 requirements) with CMR specific ones that of course match

both as required by SCMR guidelines. Research/Commercial Segment

software should be inserted in the table together with other

commercial software as Pie CAAS

(https://www.piemedicalimaging.com/product/mr-solutions/tissue-characterisation). 

The compatibility with different OS should be indicated, as one limitation

of the proposed software is the requirement of MacOS system.

 We thank the reviewer for this recommendation. We have included

 Segment CMR and CAAS MR Solutions to the software comparison. In

 addition, we have included a column “Target Platform” in Table 1. We

 have added comments on the limitation to deploy CMRSegTools on a

 different operative system.

 The Table 1 has been completed.

 And a paragraph added to the Discussion section:

 CMRSegTools is therefore a contribution to enable reproducible

 research by assembling the most used methods for infarct segmentation

 in a suitable workflow executed in a single and widely available

 environment: Osirix/Horos DICOM viewers. This platform was chosen

 because it provides an Application Programming Interface (API) that

 grants access to the data management, user interaction and

 visualisation functionalities. It was also chosen because it has an

 open-source licence. However, OsiriX/Horos is a macOS-only software,

 which limits its deployment on platforms based on a different

 operative system.

Reviewer #2

This study presents a open-source software for myocardial infarction

segmentation in CMR images. There are some concerns:

1. It is better to brief introduce the licenses shown in Table 1 for

clarifying the difference in different software.

 We thank the reviewer for this recommendation. We have added a brief

 overview of licences in the Introduction:

 In general, commercial software is a product designed to be marketed

 under a utilisation licence. The user must pay for a binary version

 which must be used as described by the licence. There is no access to

 the source code of the commercial product (closed-source). Freeware

 software, is free of charge but closed-source. In open-source

 software, the user has access to the source code and the different

 licences (i.e. BSD, GNU GPL, etc.) provide the rights to copy, modify,

 redistribute the code, as well as build and release binary versions

 with commercial purposes. [11]. Open-source software promotes open

 collaboration and research communities have been attracted to this

 innovation methodology [12].

 The following sentence has been added to the Discussion section:

 In this direction, CMRSegTools source code has been released under the

 CeCILL licence (French free software license compatible with GPL) that

 grants users the right to copy, modify, and distribute new versions of

 the plug-in.

2. Is there any comparison of BEAS with other myocardial contour

segmentation ? If not, please explain the reason.

 We thank the reviewer for this important remark.

 In CMRSegTools plug-in section, the paragraph below has been added:

 BEAS was originally developed for automatic segmentation of 3D

 cine-loop echocardiographic images to provide an efficient, fast and

 accurate solution for quantification of the main left ventricular

 volumetric indices used in clinical routine. In addition, the method

 was adapted for 3D+time CMR data sets (acquired by a cine steady state

 free precession sequence, SSFP) and benchmarked against the data sets

 available from the MICCAI 2009 Cardiac MR Left Ventricle Segmentation

 Challenge [33]. The benchmark demonstrated the technique to be robust,

 efficient and fast in terms of accuracy and computational load, which

 makes the BEAS algorithm suitable for a clinical practice [32].

3. It is better to cite more studies in cardiac and myocardial

infarction segmentation. 1) Direct delineation of myocardial

infarction without contrast agents using a joint motion feature

learning architecture, Medical Image Analysis; 2) Multi-level semantic

adaptation for few-shot segmentation on cardiac image sequences,

Medical Image Analysis.

 We thank the reviewer for the observation and references. We have

 included Xu C., et al. 2018 and Guo S. et al., 2021 in our

 bibliographical revision and discussion.

 In the Discussion section we added,

 At present, there are no automated algorithms that can reliably

 distinguish the bright LV cavity from the bright endocardial border of

 the infarct. This challenge is being addressed by new segmentation

 methods (including non-LGE images) based on ML [5,20, 47, 48].

 In the References, we added

 47. Xu C, Xu L, Gao Z, Zhao S, Zhang H, Zhang Y, et al. Direct

 delineation of myocardial infarction without contrast agents using a

 joint motion feature learning architecture. Medical image

 analysis. 2018;50:82–94.

 48. Guo S, Xu L, Feng C, Xiong H, Gao Z, Zhang H. Multi-level semantic

 adaptation for few-shot segmentation on cardiac image

 sequences. Medical Image Analysis. 2021;73:102170.

4. Please briefly discuss the limitation of this software

 In the Discussion section, the following comment has been added:

 CMRSegTools is therefore a contribution to enable reproducible

 research by assembling the most used methods for infarct segmentation

 in a suitable workflow executed in a single and widely available

 environment: Osirix/Horos DICOM viewers. This platform was chosen

 because it provides an Application Programming Interface (API) that

 grants access to the data management, user interaction and

 visualisation functionalities. It was also chosen because it has an

 open-source licence. However, OsiriX/Horos is a macOS-only software,

 which limits its deployment on platforms based on a different

 operative system.

5. A lot of grammatical errors.

 We have thoroughly reviewed and improved the grammar of the article.

6. Please present a figure to show the modules and their connections

in this software.

 We thank the reviewer for this recommendation. We added the diagram

 below presenting the main components and package dependencies in the

 CMRSegTools plug-in section:

 Figure 1 shows the main components of the plug-in and its integration

 within the OsiriX/Horos environment.

---

## [Decision Letter · Decision Letter 1]

16 Aug 2022

PONE-D-22-07768R1CMRSegTools: an open-source software enabling reproducible research in segmentation of acute myocardial infarct in CMR imagesPLOS ONE

Dear Dr. Romero R.,

Thank you for submitting your manuscript to PLOS ONE. After careful consideration, we feel that it has merit but does not fully meet PLOS ONE’s publication criteria as it currently stands. Therefore, we invite you to submit a revised version of the manuscript that addresses the points raised during the review process.

We look forward to receiving your revised manuscript.

Kind regards,

Kumaradevan Punithakumar

Academic Editor

PLOS ONE

Journal Requirements:

Reviewers' comments:

Reviewer's Responses to Questions

**Comments to the Author**

1. If the authors have adequately addressed your comments raised in a previous round of review and you feel that this manuscript is now acceptable for publication, you may indicate that here to bypass the “Comments to the Author” section, enter your conflict of interest statement in the “Confidential to Editor” section, and submit your "Accept" recommendation.

Reviewer #1: (No Response)

2. Is the manuscript technically sound, and do the data support the conclusions?

Reviewer #1: Yes

3. Has the statistical analysis been performed appropriately and rigorously? 

Reviewer #1: N/A

4. Have the authors made all data underlying the findings in their manuscript fully available?

Reviewer #1: No

5. Is the manuscript presented in an intelligible fashion and written in standard English?

Reviewer #1: Yes

6. Review Comments to the Author

Reviewer #1: Authors have addressed my requests. I have some very minor comments:

radiology expert experience should be quantified (e.g. > 10 years)

Reference 25 should be updated (pages 32(7):4340-4351)

7. PLOS authors have the option to publish the peer review history of their article (what does this mean?). If published, this will include your full peer review and any attached files.

Reviewer #1: No

---

## [Author Response · Author response to Decision Letter 1]

19 Aug 2022

Response to reviewers

Reviewer #1: Authors have addressed my requests. I have some very

minor comments:

1. radiology expert experience should be quantified (e.g. > 10 years)

 We thank the reviewer for this important remark.

 The following paragraph has been edited in the section Radiology

 expert segmentation vs automated segmentation methods:

 An example of the manual segmentation made by an expert radiologist

 (P.C. with 20 years of experience in cardiovascular imaging) against

 each one of the methods within CMRSegTools on a patient with MVO is

 presented in figures 7 and 8.

2. Reference 25 should be updated (pages 32(7):4340-4351)

 We thank the reviewer for the observation. The reference has been

 updated and following the PLOS One reference style the reference 25 is

 written as follows:

 25. Zeilinger MG, Kunze KP, Munoz C, Neji R, Schmidt M, Croisille P,

 et al. Non-rigid motion-corrected free-breathing 3D myocardial Dixon

 LGE imaging in a clinical setting. European

 Radiology. 2022;32. doi:10.1007/s00330-022-08560-6.

---

## [Editor Report · Decision Letter 2]

30 Aug 2022

CMRSegTools: an open-source software enabling reproducible research in segmentation of acute myocardial infarct in CMR images

PONE-D-22-07768R2

Dear Dr. Romero R.,

We’re pleased to inform you that your manuscript has been judged scientifically suitable for publication and will be formally accepted for publication once it meets all outstanding technical requirements.

Kind regards,

Kumaradevan Punithakumar

Academic Editor

PLOS ONE

---

## [Editor Report · Acceptance letter]

5 Sep 2022

PONE-D-22-07768R2 

CMRSegTools: an open-source software enabling reproducible research in segmentation of acute myocardial infarct in CMR images 

Dear Dr. Romero R.:

I'm pleased to inform you that your manuscript has been deemed suitable for publication in PLOS ONE. Congratulations! Your manuscript is now with our production department. 

Kind regards, 

on behalf of

Professor Kumaradevan Punithakumar 

Academic Editor

PLOS ONE